# Improvement of Malicious Software Detection Accuracy through Genetic Programming Symbolic Classifier with Application of Dataset Oversampling Techniques

Nikola Anđelić *,†, Sandi Baressi Šegota †, and Zlatan Car

Faculty of Engineering, University of Rijeka, Vukovarska 58, 51000 Rijeka, Croatia;
sbaressisegota@riteh.hr (S.B.Š.); car@riteh.hr (Z.C.)
* Correspondence: nandelic@riteh.hr
† These authors contributed equally to this work.

**Abstract:** Malware detection using hybrid features, combining binary and hexadecimal analysis with DLL calls, is crucial for leveraging the strengths of both static and dynamic analysis methods. Artificial intelligence (AI) enhances this process by enabling automated pattern recognition, anomaly detection, and continuous learning, allowing security systems to adapt to evolving threats and identify complex, polymorphic malware that may exhibit varied behaviors. This synergy of hybrid features with AI empowers malware detection systems to efficiently and proactively identify and respond to sophisticated cyber threats in real time. In this paper, the genetic programming symbolic classifier (GPSC) algorithm was applied to the publicly available dataset to obtain symbolic expressions (SEs) that could detect the malware software with high classification performance. The initial problem with the dataset was a high imbalance between class samples, so various oversampling techniques were utilized to obtain balanced dataset variations on which GPSC was applied. To find the optimal combination of GPSC hyperparameter values, the random hyperparameter value search method (RHVS) was developed and applied to obtain SEs with high classification accuracy. The GPSC was trained with five-fold cross-validation (5FCV) to obtain a robust set of SEs on each dataset variation. To choose the best SEs, several evaluation metrics were used, i.e., the length and depth of SEs, accuracy score (ACC), area under receiver operating characteristic curve (AUC), precision, recall, f1-score, and confusion matrix. The best-obtained SEs are applied on the original imbalanced dataset to see if the classification performance is the same as it was on balanced dataset variations. The results of the investigation showed that the proposed method generated SEs with high classification accuracy (0.9962) in malware software detection.

**Keywords:** genetic programming symbolic classifier; 5-fold cross-validation; malware software detection; oversampling techniques; random hyperparameter value search method





## 1. Introduction

Detecting malicious Windows executable files is of paramount importance in the realm of cybersecurity, because these files are often the primary carriers of a wide array of damaging threats, ranging from traditional viruses and trojans to sophisticated ransomware, spyware, and advanced persistent threats [1,2]. Timely and accurate detection is a vital defense mechanism against these malicious entities that can infiltrate systems, compromise data integrity, and disrupt critical operations. These files are often vehicles for cybercriminals seeking to steal sensitive information, extort financial gains, or exploit system vulnerabilities [3]. Effective detection not only shields individual users and organizations from the dire consequences of these threats, such as data breaches and financial losses, but also plays a pivotal role in safeguarding the broader digital landscape [4]. It upholds trust, reliability, and the security of interconnected systems, preventing cyberattacks and data

breaches from spreading across the digital ecosystem and ensuring the integrity of both personal and business computing environments.

Detection methods often involve advanced analysis techniques, including binary hexadecimal inspection and DLL (Dynamic Link Library) calls analysis [5]. In binary hexadecimal analysis, cybersecurity experts examine the hexadecimal representation of executable files to identify patterns or signatures associated with known malware [6]. They also scrutinize the file structure for anomalies. DLL calls analysis, on the other hand, focuses on monitoring the behavior of executables during runtime, observing the DLLs they load and the functions they call. Unusual DLL activity, API hooking, and behavioral anomalies are key indicators of potential malware [7].

The binary hexadecimal inspection consists of pattern recognition and file structure analysis. In the pattern recognition, the malware analysts often rely on identification of specific patterns and signatures within the hexadecimal representation of binary files [8]. These patterns may include unique byte sequences, headers, or other indicators that are associated with known malware strains. Antivirus and security software use signature-based detection to match these patterns against a database of known malware signatures. In file structure analysis, the structure of executable files such as portable executable format in Windows is examined in hexadecimal to understand its components [9]. Anomalies in the file structure, such as unexpected or suspicious sections, may indicate the presence of malware. This analysis helps in differentiating between legitimate and potentially malicious executables.

In dynamic analysis, malware often loads DLLs during runtime to execute various functions. This process involves monitoring the executable's behavior, keeping track of loaded DLLs, and observing the functions it calls. Security analysts scrutinize the runtime for abnormal DLL activity, unexpected dependencies, or any unusual behavior indicative of malicious intent. Additionally, API hooking detection [10] is employed to identify instances where malware intercepts and modifies legitimate API calls to achieve malicious goals. Security tools detect these activities by monitoring API calls and identifying deviations from expected behavior, such as unauthorized modifications to system functions or the redirection of API calls. Furthermore, behavioral analysis, a dynamic approach, is essential for detecting malware by examining an executable's behavior, particularly its interactions with DLLs. Unusual DLL calls, like unauthorized network connection attempts or alterations to system settings, serve as red flags during this real-time analysis, offering insights beyond static analysis reliant on file signatures.

While these techniques provide insights into malware behavior, modern security solutions employ a holistic approach, combining signature-based detection, heuristic analysis, behavior analysis, and machine learning. Automated tools and antivirus software play crucial roles in implementing these detection methods and protecting systems from evolving and sophisticated malware threats.

Over the years, several research papers have shown excellent applications of artificial intelligence in malware executable/code detection. Malware detection was investigated in [11] using a random forest classifier (RFC) and deep neural networks (DNN) with two, four, and seven layers, respectively. The highest classification accuracy was achieved with RFC (99.78%). A web-based framework was developed in [12] for the detection of malware from Android devices. This research included support vector machine (SVM), RFC, multi-layer perceptron (MLP), logistic regression (LR), Bayesian network (BN), Adaboost (AB), decision tree (DT), k-nearest neighbors (KNN), DNN, self-organizing map (SOM), K-mean, farthest first clustering (FF), filtered clustering (FC), density-based clustering (DB), J48 + YATSI (Y-J48), SMO + YATSI (Y-SMO), MLP + YATSI (Y-MLP), best training ensemble approach (BTE), majority voting ensemble approach (MVE), and nonlinear ensemble decision tree forest approach (NDTF). The investigation showed that the model which consisted of NDTF, Y-MLP, DNN, and FF clustering can detect 98.8% malware from real-world applications. The LR, NB, KNN, DT, AB, RFC, SVM, Convolutional Neural Networks, MLP + SVM, CNN + long short-term memory (LSTM) have been used in [13] to

detect malware. Each ML algorithm was trained using 10-fold cross-validation. The highest classification accuracy of 98.8% was achieved with two-layer CNN + LSTM. The malware detection using ML-based analysis of virtual memory access pattern was performed in [14]. In this research, the LR, SVM, and RFC have been used to classify kernel rootkits and memory corruption attacks on user programs. The highest accuracy achieved was 100% using the RFC algorithm. The accuracy results are summarized in Table 1.

**Table 1.** The summarized results in malware detection/classification reported in other research papers.

| Reference | ML Algorithms | Accuracy (%) |
| --- | --- | --- |
| [11] | RFC, DNN-2,4,7 | 99.78 |
| [12] | SVM, RFC, MLP, LR, BN, AB, DT, KNN, DNN, SOM, FF, FC, DB, Y-J48, Y-SMO, Y-MLP, BTE, and MVE | 98.80 |
| [13] | LR, NB, KNN, DT, AB, RFC, SVM, CNN, MLP + SVM, and CNN + LSTM | 98.80 |
| [14] | LR, SVM, RFC | 100.00 |

Although the results of previous research shown in Table 1 are very high the main problem is the inability to transform these trained models into mathematical equations that can be easily executed. So in this research, the idea is to apply the genetic programming symbolic classifier to a publicly available dataset [15] to obtain SEs that can be easily implemented and used in the detection of malware executable (software). The dataset contains two classes, i.e., malicious and non-malicious software. The main problem with the dataset is that the dataset is imbalanced, which means that there is a large difference between class samples. However, due to the small number of dataset samples (373), the oversampling methods will be applied to create balanced variations of the original dataset. Since GPSC has a large number of hyperparameters the random hyperparameter values search method will be developed and applied to find the optimal combination of hyperparameters, using which the SEs with high classification accuracy are obtained. The GPSC will be trained using five-fold cross-validation to obtain a robust set of SEs with high accuracy obtained on each dataset variation.

Based on the detailed literature overview and the disadvantages of used approaches, the following questions arise:

- Is it possible to apply various oversampling techniques to obtain the balanced dataset variations of the original imbalanced datasets?
- Is it possible to apply the GPSC algorithm to obtain SEs that could detect malicious software with high classification performance?
- Is it possible to find the optimal combination of GPSC hyperparameters by developing and applying the random hyperparameter value search method?
- Is it possible to obtain a robust set of SEs by training the GPSC using five-fold cross-validation?
- Is it possible to achieve the same classification performance by combining all SEs obtained on balanced dataset variations on the original imbalanced dataset?

This paper consists of the following sections:

- Materials and Methods—in this section the research methodology, dataset, utilized data preprocessing and machine learning methods, evaluation metrics, and training procedure are described.
- Results—the results of the conducted research are presented by showing optimal hyperparameter values using which SEs are obtained with high classification accuracy. The obtained SEs on each dataset variation are then tested on the original dataset to see their performance.

- Discussion—contains discussion on obtained results presented in the previous section
- Conclusions—in this section, the conclusions are presented based on the hypothesis given in this section.

Additionally, the appendix section contains information on how to download the SEs from the GitHub repository as well as procedures on how to use these SEs.

## 2. Materials and Methods

This section consists of the research methodology, a short dataset description used in this research, a description of used oversampling techniques, GPSC with RHVS, training/testing procedure, and used evaluation metrics.

### 2.1. Research Methodology

The graphical representation of the research methodology is shown in Figure 1.

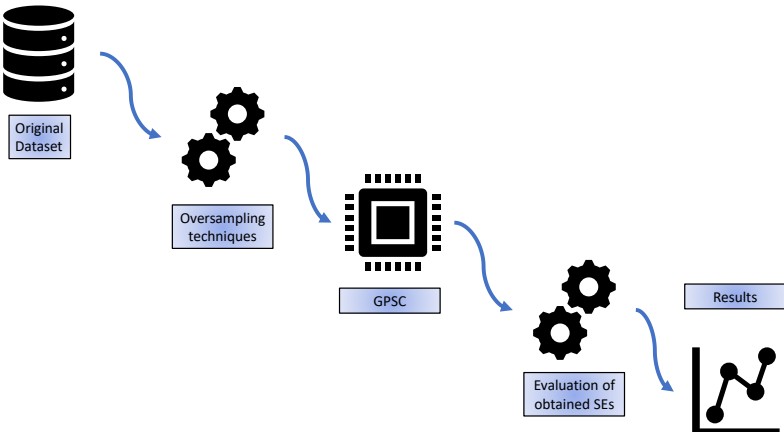

**Figure 1.** The flowchart of research methodology.

From Figure 1 it can be seen that the research methodology consists of the following steps:

1. Original dataset—the initial investigation of the dataset is performed.
2. Oversampling techniques—the idea is to apply various oversampling techniques to obtain balanced dataset variations from the original imbalanced dataset,
3. GPSC—application of the GPSC algorithm with RHVS method on each dataset variation. For training of GPSC, the 5FCV method was used and if all evaluation metrics are greater than 0.99 then the final testing of obtained SEs is performed.
4. Evaluation of obtained SEs—after GPSC + RHVS + 5FCV is applied on all dataset variations, all best SEs will be combined and evaluated on the original imbalanced dataset to see if the classification performance is the same/similar to the values obtained on the original imbalanced dataset.
5. Results—presentation of obtained results on both balanced dataset variations and imbalanced original dataset.

### 2.2. Dataset Description

In this paper, the publicly available dataset at Kaggle [15] was used. This dataset consists of 531 input variables and one output target variable (malicious/non-malicious). According to the dataset description in [15], the dataset contains the features extracted from malicious and non-malicious Windows executable files. The dataset input variables are hybrid features that combine binary hexadecimal and Dynamic Link Library (DLL) calls from Windows executables.

Hybrid features in the context of Windows executables refer to combining information from multiple sources or aspects of an executable file to create a more comprehensive and informative feature set for analysis and detection. Hybrid features combine binary hexadecimal data with DLL (Dynamic Link Library) calls.

The binary hexadecimal data typically involve extracting and encoding the binary content of an executable file into hexadecimal format. Each byte of the file is represented by two hexadecimal digits. This representation can capture the file's raw data, including the instructions and data it contains. Analyzing binary hexadecimal data can help identify patterns, signatures, or anomalies within the file, which may be indicative of malicious behavior.

The DLLs are files that contain code and data that multiple programs can use simultaneously. Windows executables often rely on DLLs for various functions, and these interactions can provide valuable insights into the executable's behavior. Monitoring DLL calls can help detect suspicious or unauthorized activity, such as attempts to load or interact with malicious DLLs.

The idea behind combining binary hexadecimal data with DLL calls as hybrid features is to create a more holistic view of the executable's behavior and content. By analyzing both the low-level binary structure (hexadecimal data) and the dynamic runtime behavior (DLL calls), security researchers and malware analysts can better understand the file's purpose, potential risks, and any malicious intent.

These hybrid features can be used as inputs for machine learning models, signature-based detection systems, or behavioral analysis tools to improve the accuracy of detecting malicious or suspicious Windows executables. By considering both static and dynamic aspects of the file, security systems can better adapt to the evolving landscape of malware and cyber threats.

The first problem with dataset statistical analysis is that the dataset contains 531 input features/variables. The creation and representation of a correlation matrix is almost impossible so another approach is considered. The correlation analysis used in this research is Pearson's correlation analysis. The correlation between any two features/variables in the dataset can be in the range from −1 to 1. Where −1 presents the perfect negative, while the other perfect positive correlation. If the correlation between two variables is negative, i.e., −1 this indicates that if the value of one variable increases the value of the other decreases. If the value is 1, then if the value of one variable increases the value of the other also increases. The worst correlation between variables is 0, which means that one variable does not have any effect on the other variable.

In this investigation, the idea is to see if any of those input variables correlate with the target variable higher than 0.5. In Figure 2, the correlation between input variables and output variable is shown.

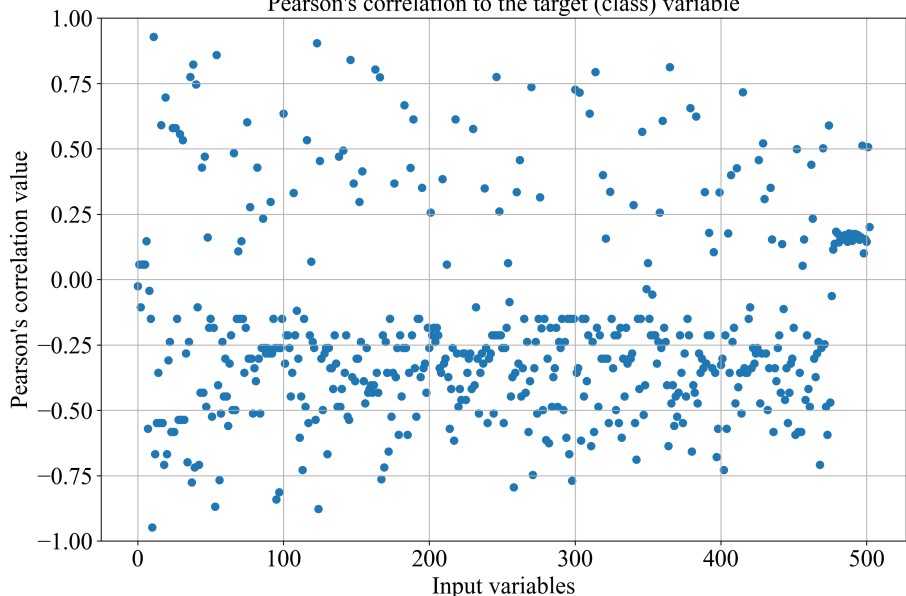

**Figure 2.** The results of Pearson correlation analysis of all input variables to the target variable.

It should be noted that after performing the Pearson's correlation analysis, some of the variables had "NaN" values with the output variable. These input variables in the original dataset are named as F_2, F_6, F_8, F_10, F_12, F_14, F_16, F_18, F_22, F_28, F_32, F_34, F_88, F_90, F_96, F_184, F_204, F_230, F_232, F_450, F_456, F_458, F_460, F_482, F_486, F_488, F_490, and F_492. The reason why these values had "NaN" values in the correlation analysis is that for all the samples in the dataset, the values of these variables are equal to 0. As seen from Figure 2, the majority of input variables have a negative correlation in the range from −0.75 up to 0.2 with the target variable. The small part of the input variables has a correlation value with the target variable around 0 which means they are almost independent variables. There is also a small part of input variables with a positive correlation to the target variable.

Although there are small parts of input variables with NaN correlation values to the target variable the idea is to see which input variables will be included in the SEs after the application of GPSC, so all input variables were used in this investigation. One of the major problems with this dataset is the large number of malicious and small number of non-malicious software. Using LabelEncoder the target variable was transformed to malicious software labeled as 1 and non-malicious labeled as 0. Further analysis showed that there are 301 malicious samples and 72 non-malicious samples, so when comparing these two classes it can be noticed that the dataset is highly imbalanced. It is well known that the performance of the majority of AI algorithms can suffer from training on imbalanced datasets. To overcome this problem various oversampling techniques will be applied to synthetically balance the dataset as much as possible. Figure 3 shows the class samples in bar form.

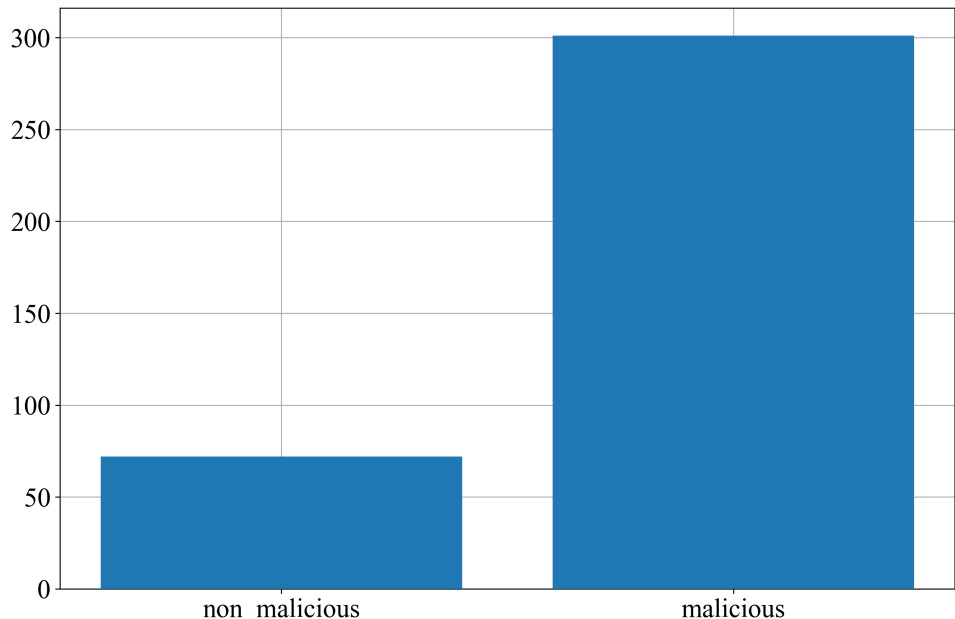

**Figure 3.** The number of samples per class is displayed in bar form.

### 2.3. Oversampling Techniques

In this research 5 different oversampling techniques were chosen, i.e., ADASYN, BorderlineSMOTE, KMeanSMOTE, SMOTE, and SVMSMOTE. These techniques were selected for three main reasons: dealing with a small dataset, ensuring faster execution, and eliminating the need for parameter tuning. The dataset used in this study is limited, with 373 samples in total, 72 in the minority (non-malicious) class, and 301 in the majority (malicious) class. Oversampling techniques are preferred for quickly balancing datasets compared to under-sampling techniques like AllKNN and Edited Nearest Neighbors. For instance, AllKNN, an undersampling method, identifies and removes instances close to their k-nearest neighbors in the majority class, which can be time-consuming regardless of

the sample size. Another advantage of oversampling is the absence of parameter tuning, as these techniques often achieve balanced datasets without any tuning. In contrast, undersampling techniques may require multiple executions with different parameters to achieve class balance. The minority class is the class that will be synthetically oversampled using the oversampling technique while the majority class will remain unchanged. In this investigation, the minority class is the non-malicious class while the majority class is the malicious class.

Synthetic Minority Over-sampling Technique (SMOTE) is lauded for its simplicity and effectiveness in mitigating class imbalance by generating synthetic samples along line segments connecting minority class instances [16]. However, its drawback lies in its potential to introduce noisy samples in regions of overlapping classes, which may impact model generalization.

Adaptive Synthetic Sampling (ADASYN) addresses SMOTE's limitations by adaptively generating more synthetic samples for minority class instances that are harder to learn [17]. This adaptability is beneficial, but the method comes at a higher computational cost compared to basic SMOTE, making it less suitable for large datasets.

BorderlineSMOTE focuses on generating synthetic samples for instances near the decision boundary, contributing to a reduction in noise compared to basic SMOTE [18]. Nonetheless, its performance may suffer when faced with highly complex decision boundaries.

KMeansSMOTE introduces structure to synthetic sample generation by combining KMeans clustering with SMOTE [19]. While it attempts to create samples in a more controlled manner, it is sensitive to the choice of the K parameter and may not perform well with non-convex clusters.

SVMSMOTE utilizes support vector machines to identify hard-to-learn instances and tailors synthetic sample generation accordingly [20]. It excels in capturing complexities in decision boundaries but is computationally more expensive than basic SMOTE and may require the careful tuning of parameters for optimal performance. Choosing the most suitable method depends on the specific characteristics of the dataset and the desired trade-off between computational cost and oversampling effectiveness.

The number of samples generated after each application of the previously described oversampling technique is listed in Table 2.

**Table 2.** Results of the application of different oversampling techniques on the original dataset.

| Dataset Variation | Class 0 Samples | Class 1 Samples | Total Number of Samples |
|---|---|---|---|
| Original dataset | 72 | 301 | 373 |
| ADASYN | 300 | 301 | 601 |
| BorderlineSMOTE | 301 | 301 | 602 |
| KMeansSMOTE | 304 | 301 | 605 |
| SMOTE | 301 | 301 | 602 |
| SVMSMOTE | 301 | 301 | 602 |

As seen from Table 2 the dataset was successfully balanced with the application of different oversampling techniques. However, there are small imbalances in the generated dataset variation. In the case of BorderelineSMOTE, SMOTE, and SVMSMOTE the generated datasets are perfectly balanced, i.e., an equal number of samples in each class. However, in the case of ADASYN the minority class (class 0) has one sample less than the majority class (class_1), and in the case of KMeansSMOTE the minority class has three samples more than the majority class.

### 2.4. Genetic Programming Symbolic Classifier with Random Hyperparameter Value Selection

The genetic programming symbolic classification algorithm [21] begins by creating the initial population of mathematical formulas represented as the three forms and these population members are unfit for a particular task. Their initial poor performance means that they have poor classification performance. Then, with the application of genetic

operators from generation to generation, the population members evolve, and after a consecutive number of generations, the best population member is obtained. To find the optimal hyperparameters of GPSC the random hyperparameter search method was used to find the optimal combination of hyperparameters, using which the best values of used evaluation metric values are obtained. The random hyperparameter value search method is developed by performing an initial investigation of the GPSC algorithm with the boundary values of each hyperparameter. In the RHVS method, the following GPSC hyperparameter values were searched i.e.,

- PopSize—the size of the population.
- numGen—the maximum number of generations. This hyperparameter is one of the stopping criteria (in this investigation it was the dominating one).
- initDepth—the initial depth of population members when represented in the tree form.
- TourSize—the size of tournament selection. How many members will be randomly picked from the population to compete in tournament selection?
- Crossover—the probability of performing the crossover operation on the tournament winner. For this genetic operation, two tournament winners are needed and the subtree from one tournament winner is used and replaces the randomly selected subtree of the second tournament winner.
- subtreeMute—the probability of performing subtree mutation. This type of mutation begins by random selection of the subtree on the tournament winner. The second step is to create a subtree at random using constants, variables, and mathematical functions to produce a new subtree, thus creating a new population member.
- pointMute—the probability of performing the point mutation. It begins with a random selection of the tournament winner nodes which have to be replaced. The variables are replaced with other variables, constants with other constants, and mathematical functions with other mathematical functions. However, when these functions are replaced the new function must have the same number of variables as the original function.
- hoistMute—the probability of performing the hoist mutation of the tournament winner. This genetic operation begins by selecting a random subtree from the tournament winner and then selecting a random node (point) on that subtree. The randomly selected node replaces the entire randomly selected subtree.
- constRange—the range of constants arbitrarily defined.
- max samples—the max number of training samples used to evaluate population members in each generation. If the value is set to 1 then the raw fitness value is not visible during the GPSC execution, so the value is set between 0.99 and 1.
- ParsCoeff—this hyperparameter is very important since it regulates the use of the parsimony pressure method. This method is crucial during the GPSC execution to prevent the bloat phenomenon, i.e., the rise in the size of population members without any benefit to fitness function value. The parsimony pressure method is applied during the tournament selection in which large population members in terms of size are made less favorable for selection by multiplying their fitness value with the parsimony coefficient.

The mathematical functions used in GPSC to develop the population members consist of $+$, $-$, $\cdot$, $/$, $\sqrt{}$, $\sqrt[3]{}$, $\log$, $\log_2$, $\log_{10}$, $\sin$, $\cos$, and $\tan$. However, the mathematical functions such as division, sqrt, natural logarithm, and logarithm with bases 2 and 10 have been modified to avoid the generation of not a number or infinite values during the GPSC execution. These modifications are given in the Appendix A.1 section.

It should be noted that the sum of all genetic operators must be equal to 1. If the value is lower than 1 then some winners of tournament selection might enter the next generation unchanged, i.e., reproduction will occur.

To determine the quality of each population member, the fitness function is required. In the case of GPSC to determine the fitness function the following steps are required:

1. Using values of input variables calculate the output of population member,

2. Uses the generated output as input in the Sigmoid function which can be written as:

$$S(x) = \frac{1}{1 + e^{-x}} \tag{1}$$

where $x$ is the output of the population member.

3. Uses the output of the sigmoid function and the real output from the dataset to calculate the LogLoss value. The LogLoss according to [] is calculated using the following expression:

$$LogLoss(x) = -\frac{1}{N} \sum_{i=1}^{N} y_i \log(p(y_i)) + (1 - y_i) \log(1 - p(y_i)) \tag{2}$$

where $p(y_i)$ is the probability of 1, and $1 - p(y_i)$ is the probability of 0.

To define the boundaries of each GPSC hyperparameter, the initial boundaries were arbitrarily defined and each hyperparameter boundary was shortly tested in the GPSC algorithm. The most sensitive hyperparameter was the parsimony coefficient since its value highly influences the occurrence of the bloat phenomenon.

The ranges of all hyperparameter values used in this research with the RHVS method are shown in Table 3.

**Table 3.** The GPSC hyperparameter value ranges used in the RHVS method.

| Hyperparameter Name | Lower Boundary | Upper Boundary |
|---|---|---|
| PopSize | 1000 | 2000 |
| numGen | 200 | 300 |
| initDepth | 3 | 18 |
| TourSize | 100 | 500 |
| Crossover | 0.001 | 1 |
| subtreeMute | 0.001 | 1 |
| hoistMute | 0.001 | 1 |
| pointMute | 0.001 | 1 |
| constRange | $-10{,}000$ | 10,000 |
| max samples | 0.99 | 1 |
| ParsCoeff | $1 \times 10^{-6}$ | $1 \times 10^{-5}$ |

### 2.5. Training Procedure

The training procedure is shown in Figure 4.

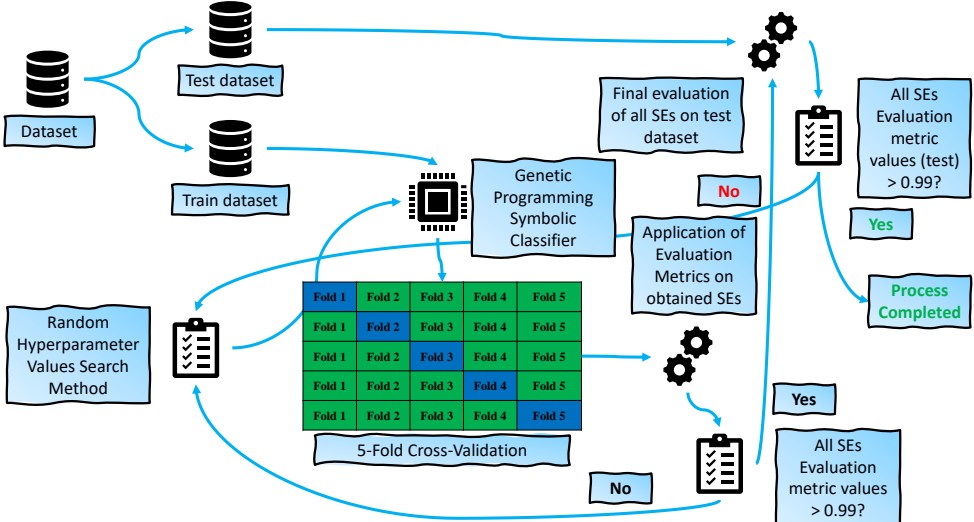

**Figure 4.** The graphical representation of the training/testing procedure used in this research.

In Figure 4, the procedure of obtaining the best SEs on each balanced dataset variation with highest classification accuracy is shown using the process of GPSC random hyper-parameter values search (RHVS) method and GPSC trained with 5-fold cross-validation (5-FCV) [22]. The process shown in Figure 4 consists of following steps:

1. The dataset (balanced dataset variation) was initially divided into training and testing datasets in a 70:30 ratio, where 70% was used for training using a 5-fold cross-validation process and the remaining 30% for testing. At this stage, the GPSC hyper-parameters are randomly chosen from the predefined ranges.

2. When the dataset is split into train and test datasets and the random hyperparameter values are chosen using the RHVS method, the training of the GPSC algorithm in the 5-FCV process can begin. In 5-FCV, the GPSC is trained 5 times, i.e., a total of 5 SEs are obtained after the 5-FCV process is completed. After the 5-FCV process is completed, the accuracy, area under the receiver operating characteristics curve, precision, recall, and f1-score values are obtained for training and validation folds. Then the mean and standard deviation of the previously mentioned evaluation metrics are calculated.

3. All mentioned mean evaluation metric values have to be greater than 0.99 to be considered as the set of SEs with acceptable classification performance. If one of the evaluation metrics is lower than 0.99 in value, all the SEs are neglected and the process starts from the beginning, where random hyperparameters are randomly selected and GPSC training using the 5-FCV process is repeated. However, if all evaluation metric values are higher than 0.99, the process continues to the testing phase.

4. In the testing phase, the remaining 30% of the dataset is used on all 5 SEs obtained from the previous step. Using the testing set in these 5SEs, the output is generated and compared to the real output to calculate the evaluation metric values. If all evaluation metric values are higher than 0.99, then the process is completed. However, if one evaluation metric value is lower than 0.99 the process starts from beginning by randomly selecting the hyperparameter values.

### 2.6. Evaluation Metrics

In this research, the accuracy score, the area under the receiver operating characteristics, precision score, recall score, and f1-score were used. Additionally, the confusion matrix was used to see the number of correct/incorrect predictions.

The accuracy score [23] provides information about how many predictions were true. For binary classification problems, the accuracy score can be written as:

$$ACC = \frac{TP + TN}{TP + TN + FP + FN},\tag{3}$$

where $TP, TN, FP$, and $FN$ are true positives, true negatives, false positives, and false negatives, respectively. The area under the receiver operating characteristics curve [24] is a metric used to evaluate the performance of binary classification models. The ROC curve is a graphical representation of the model's performance across different thresholds for classifying the positive and negative classes. The AUC value quantifies the overall discriminative power of the model. To calculate the area under the ROC curve, first, the predictions are generated from the binary classification model. These scores represent the likelihood of an instance belonging to the positive class. The next step is to calculate the ROC curve using the true labels and predicted scores. The ROC curve plots the True Positive Rate (TPR) against the False Positive Rate (FPR) at various threshold levels. Once the ROC curve is obtained, the AUC can be calculated by computing the area under this curve.

The precision score [25] answers what portion of identifications was correct. For binary problems, the precision score is calculated as:

$$Precision = \frac{TP}{TP + FP}.\tag{4}$$

The recall score [25] answers what proportion of actual positives was identified correctly. The recall score can be calculated using the expression:

$$Recall = \frac{TP}{TP + FN}.$$ (5)

The F1-score is the harmonic balance between precision and recall and is calculated as:

$$F1 - Score = \frac{2 \cdot PRecision \cdot Recall}{Precision + Recall}.$$ (6)

A confusion matrix [26] is a concise tabular representation used in the realm of ML and statistics to assess the performance of a classification model. It provides a breakdown of how the model's predictions align with the actual true labels, summarizing the number of true positives (correct positive predictions), true negatives (correct negative predictions), false positives (incorrect positive predictions), and false negatives (incorrect negative predictions). This matrix serves as a crucial tool for evaluating a model's classification accuracy, enabling the computation of various performance metrics such as accuracy, precision, recall, specificity, and F1-score, which collectively measure the model's ability to correctly classify instances and identify areas for potential improvement in classification tasks.

## 3. Results

The results section contains two subsections i.e., the results obtained on the oversampled variations of the dataset, and the detection performance of the previously obtained symbolic expressions on the original dataset.

### 3.1. The Results Obtained on the Oversampled Datasets

After the application of each oversampling technique, a new dataset (Balanced) variation is created. On each dataset variation, the GPSC algorithm with RHVS method and 5FCV were applied. The results consist of two segments, i.e., the presentation of optimal hyperparameters, and classification performance in terms of evaluation metric values. The optimal GPSC hyperparameters for each dataset variation are listed in Table 4.

**Table 4.** The GPSC optimal hyperparameters are used in each dataset variation to obtain SEs with high classification performance.

| Dataset Name | GPSC Hyperparameters |
|:---:|:---:|
| ADASYN | 1392, 175, 119, (7, 10), 0.0226, 0.96, 0.0093, 0.0061, $3 \times 10^{-6}$, 0.995, $(-9425.21, 4890.24)$, $1.16 \times 10^{-6}$ |
| BorderlineSMOTE | 1829, 168, 115, (3, 9), 0.0012, 0.96, 0.021, 0.01, $7 \times 10^{-6}$, 0.993, $(-9259.86, 3343.17)$, $7.59 \times 10^{-6}$ |
| KMeansSMOTE | 1943, 192, 120, (5, 12), 0.018, 0.96, 0.0047, 0.012, $4 \times 10^{-6}$, 0.99, $(-1423.02, 3973.33)$, $8.73 \times 10^{-6}$ |
| SMOTE | 1804, 187, 493, (4, 11), 0.031, 0.96, 0.004, 0.0032, $1 \times 10^{-5}$, 0.996, $(-4652.34, 3243.75)$, $3.192 \times 10^{-6}$ |
| SVMSMOTE | 1331, 187, 436, (4, 10), 0.0034, 0.95, 0.0093, 0.027, $2 \times 10^{-6}$, 0.99, $(-7067.54, 16.16)$, $8.1887 \times 10^{-6}$ |

As seen from Table 4 the highest population size was used in the case of BorderlineSMOTE, KMeansSMOTE, and SMOTE. The maximum number of generations is higher than 150, i.e., for ADASYN, KMeansSMOTE, SMOTE, and SVMSMOTE all values are near the

upper boundary (Table 4. The highest size of tournament selection value was in the case of SMOTE and SVMSMOTE datasets while in the remaining three cases, the tournament size was near the 100 (lower boundary) value. The largest range of initial depth in the case of the initial population was in the case of KMeansSMOTE and SMOTE datasets. In all GPSC executions, the maximum number of generations was used as termination criteria, since the minimum value of the fitness function (stopping criteria) was never met. The most dominating genetic operator in all investigations was the subtree mutation (0.95 or higher). The largest range of constant values was used in the case of the ADASYN dataset ($-9425.21$, $4890.24$). The parsimony coefficient value in all investigations was near the lower boundary defined in Table 3. The performance of obtained SEs on each dataset variation is shown in Figure 5.

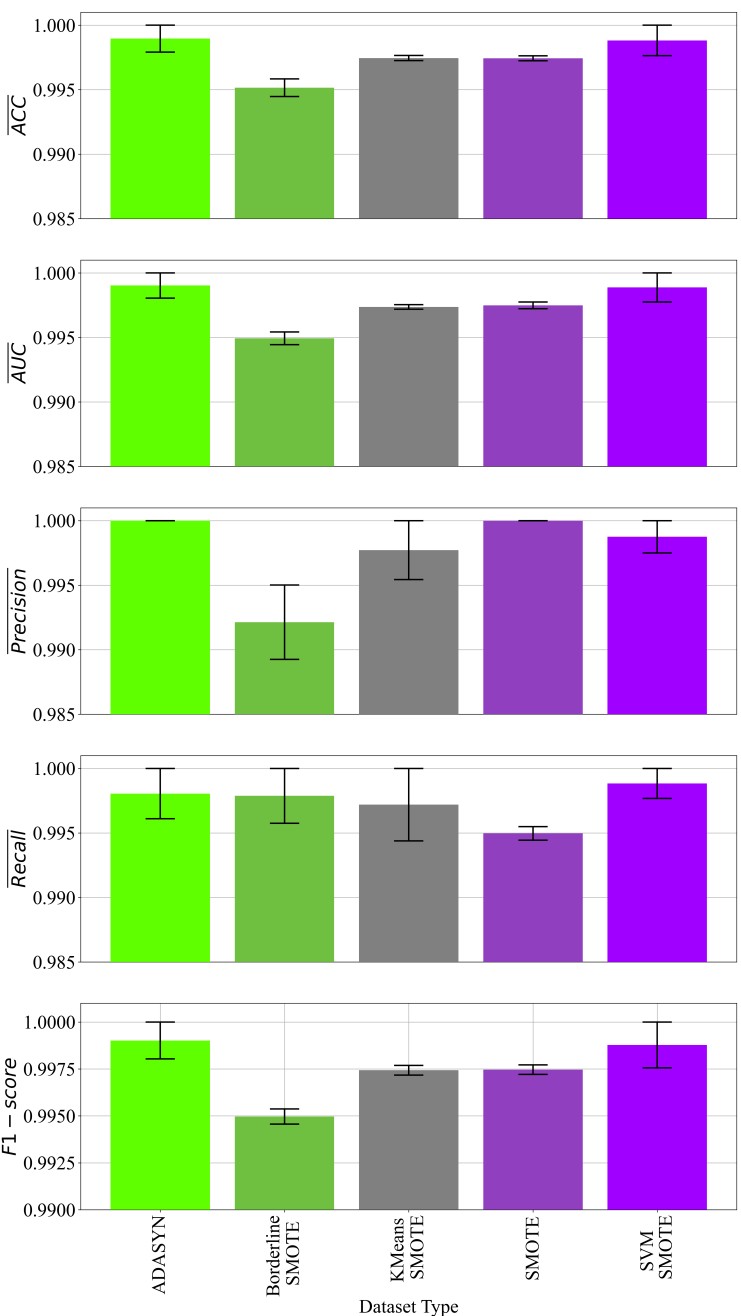

**Figure 5.** The classification performance of obtained SEs. The standard deviation ($\sigma$) values are represented as error bars.

From Figure 5, it can be noticed that all obtained SEs on every dataset variation achieved the highest classification performance. However, the highest classification performance was achieved in the case of the dataset balanced with the ADASYN technique. However, the $\sigma$ values are noticeable they are all small when compared to the achieved accuracy, which shows that overfitting did not occur. In Table 5, the size and depth of population members are listed.

**Table 5.** The depth, length, mean depth, and mean length of obtained SEs on each dataset variation.

| Dataset Type | Depth of SEs | Length of SEs | Mean Depth | Mean Length |
|---|---|---|---|---|
| ADASYN | 13/17/12/22/13 | 52/169/95/91/92 | 15.4 | 99.8 |
| BorderlineSMOTE | 17/10/9/11/11 | 131/77/28/32/96 | 11.6 | 72.8 |
| KMeansSMOTE | 13/15/15/26/11 | 116/56/70/73/33 | 16.0 | 69.6 |
| SMOTE | 14/7/11/17/6 | 83/17/108/96/10 | 11 | 62.8 |
| SVMSMOTE | 20/20/9/22/19 | 112/38/32/55/88 | 18.0 | 65.0 |

From Table 5, it can be noticed based on the mean length of obtained SEs that the largest SEs were obtained on dataset variation balanced with the ADASYN technique. The smallest set of SEs was obtained in the case of a dataset balanced with the SMOTE technique based on mean length. In terms of mean depth (largest SEs in tree form), the largest SEs were obtained in the case of the dataset balanced with the SVMSMOTE technique. The smallest mean depth was achieved with SEs generated on the dataset balanced with the SMOTE technique.

To show what one of the SEs looks like the following expression represents the shortest SE obtained on dataset variation balanced with SMOTE method.

$$
y_{SMOTE_5} = \tan\left( \tan\left( X_{19} - \frac{\max\left( X_{46}, \frac{\log(X_{365})}{\log(10)} \right)}{X_{48}} \right) \right). \tag{7}
$$

By analyzing all the best SEs, a total of 303 input variables to calculate the output were found, and these variables are: $X_0$ $(F_1) - X_{57}$ $(F_{58})$, $X_{59}$ $(F_{60}) - X_{63}$ $(F_{64})$, $X_{65}$ $(F_{66})$, $X_{66}$ $(F_{67})$, $X_{68}$ $(F_{69})$, $X_{72}$ $(F_{73}) - X_{74}$ $(F_{75})$, $X_{78}$ $(F_{79})$, $X_{81}$ $(F_{82})$, $X_{83}$ $(F_{84})$, $X_{85}$ $(F_{86}) - X_{88}$ $(F_{89})$, $X_{90}$ $(F_{91})$, $X_{93}$ $(F_{94})$, $X_{95}$ $(F_{96})$, $X_{96}$ $(F_{97})$, $X_{99}$ $(F_{100}) - X_{101}$ $(F_{102})$, $X_{106}$ $(F_{107})$, $X_{109}$ $(F_{110})$, $X_{112}$ $(F_{113})$, $X_{114}$ $(F_{115})$, $X_{116}$ $(F_{117}) - (F_{119})$, $X_{121}$ $(F_{122}) - X_{123}$ $(F_{124})$, $X_{129}$ $(F_{130})$, $X_{131}$ $(F_{132})$, $X_{132}$ $(F_{133})$, $X_{134}$ $(F_{135})$, $X_{135}$ $(F_{136})$, $X_{138}$ $(F_{139})$, $X_{139}$ $(F_{140})$, $X_{141}$ $(F_{142}) - X_{143}$ $(F_{144})$, $X_{145}$ $(F_{146})$, $X_{149}$ $(F_{150})$, $X_{150}$ $(F_{151})$, $X_{153}$ $(F_{154})$, $X_{156}$ $(F_{157})$, $X_{159}$ $(F_{160})$, $X_{160}$ $(F_{161}) - X_{164}$ $(F_{165})$, $X_{167}$ $(F_{168})$, $X_{170}$ $(F_{171})$, $X_{171}$ $(F_{172})$, $X_{174}$ $(F_{175}) - X_{176}$ $(F_{177})$, $X_{178}$ $(F_{179}) - X_{181}$ $(F_{182})$, $X_{183}$ $(F_{184})$, $X_{188}$ $(F_{189}) - X_{192}$ $(F_{193})$, $X_{194}$ $(F_{195}) - X_{197}$ $(F_{198})$, $X_{199}$ $(F_{200})$, $X_{204}$ $(F_{205}) - X_{208}$ $(F_{209})$, $X_{210}$ $(F_{211}) - X_{215}$ $(F_{216})$, $X_{218}$ $(F_{219}) - X_{220}$ $(F_{221})$, $X_{222}$ $(F_{223})$, $X_{223}$ $(F_{224})$, $X_{225}$ $(F_{226})$, $X_{229}$ $(F_{230}) - X_{234}$ $(F_{235})$, $X_{236}$ $(F_{237})$, $X_{240}$ $(F_{241}) - X_{245}$ $(F_{246})$, $X_{248}$ $(F_{249})$, $X_{254}$ $(F_{255})$, $X_{256}$ $(F_{257})$, $X_{257}$ $(F_{258})$, $X_{259}$ $(F_{260})$, $X_{262}$ $(F_{263})$, $X_{265}$ $(F_{266}) - X_{267}$ $(F_{268})$, $X_{269}$ $(F_{270})$, $X_{273}$ $(F_{274})$, $X_{275}$ $(F_{276})$, $X_{277}$ $(F_{278})$, $X_{279}$ $(F_{280})$, $X_{280}$ $(F_{281}) - X_{282}$ $(F_{283})$, $X_{284}$ $(F_{285})$, $X_{285}$ $(F_{286})$, $X_{287}$ $(F_{288})$, $X_{290}$ $(F_{291})$, $X_{294}$ $(F_{295})$, $X_{295}$ $(F_{296})$, $X_{298}$ $(F_{299})$, $X_{301}$ $(F_{302})$, $X_{302}$ $(F_{303})$, $X_{304}$ $(F_{305})$, $X_{308}$ $(F_{309})$, $X_{310}$ $(F_{311}) - X_{312}$ $(F_{313})$, $X_{314}$ $(F_{315}) - X_{317}$ $(F_{318})$, $X_{320}$ $(F_{321})$, $X_{321}$ $(F_{322})$, $X_{325}$ $(F_{326})$, $X_{328}$ $(F_{329}) - X_{330}$ $(F_{331})$, $X_{331}$ $(F_{332})$, $X_{336}$ $(F_{337}) - X_{338}$ $(F_{339})$, $X_{340}$ $(F_{341}) - X_{342}$ $(F_{343})$, $X_{344}$ $(F_{345})$, $X_{348}$ $(F_{349}) - X_{351}$ $(F_{352})$, $X_{354}$ $(F_{355})$, $X_{355}$ $(F_{356})$, $X_{360}$ $(F_{361})$, $X_{362}$ $(F_{363})$, $X_{365}$ $(F_{366})$, $X_{368}$ $(F_{369})$, $X_{373}$ $(F_{374})$, $X_{377}$ $(F_{378})$, $X_{380}$ $(F_{381})$, $X_{381}$ $(F_{382})$, $X_{384}$ $(F_{385})$, $X_{386}$ $(F_{387}) - X_{388}$ $(F_{389})$, $X_{392}$ $(F_{393})$, $X_{393}$ $(F_{394})$, $X_{395}$ $(F_{396})$, $X_{398}$ $(F_{399})$, $X_{399}$ $(F_{400})$, $X_{401}$ $(F_{402})$, $X_{403}$ $(F_{404})$, $X_{404}$ $(F_{405})$, $X_{406}$ $(F_{407})$, $X_{408}$ $(F_{409})$, $X_{409}$ $(F_{410}) - X_{411}$ $(F_{412})$, $X_{414}$ $(F_{415})$, $X_{415}$ $(F_{416})$, $X_{419}$ $(F_{420})$, $X_{420}$ $(F_{421})$, $X_{423}$ $(F_{424}) - X_{425}$ $(F_{426})$, $X_{427}$ $(F_{428}) - X_{430}$ $(F_{431})$, $X_{433}$ $(F_{434})$, $X_{434}$ $(F_{435})$, $X_{438}$ $(F_{439}) - X_{440}$ $(F_{441})$, $X_{442}$ $(F_{443}) - X_{445}$ $(F_{446})$, $X_{448}$ $(F_{449})$, $X_{450}$ $(F_{451})$, $X_{451}$ $(F_{452})$, $X_{454}$ $(F_{455})$, $X_{455}$ $(F_{456})$, $X_{457}$ $(F_{458})$, $X_{463}$ $(F_{464}) - X_{466}$ $(F_{467})$, $X_{471}$ $(F_{472})$, $X_{473}$ $(F_{474}) - X_{476}$ $(F_{477})$, $X_{478}$ $(F_{479})$, $X_{481}$ $(F_{482})$, $X_{489}$ $(F_{490}) - X_{491}$ $(F_{492})$, $X_{495}$ $(F_{496})$, $X_{497}$ $(F_{498})$, $X_{500}$ $(F_{501}) - X_{502}$ $(F_{503})$, $X_{504}$ $(F_{505})$, $X_{506}$ $(F_{507})$, $X_{508}$ $(F_{509})$, $X_{510}$ $(F_{511}) - X_{514}$ $(F_{515})$, $X_{515}$ $(F_{516})$, $X_{523}$ $(F_{524}) - X_{526}$ $(F_{527})$,

$X_{528}$ ($F_{529}$), and $X_{530}$ ($F_{531}$). In the previous section's ("Materials and Methods") subsection "Dataset Description", it was mentioned that 28 input variables have 0 values for all the samples in the dataset. However, out of those 28 input variables, only 21 variables are in the obtained SEs (all mentioned input variables except $X_{89}$ ($F_{90}$), $X_{183}$ ($F_{184}$), $X_{449}$ ($F_{450}$), $X_{459}$ ($F_{460}$), $X_{485}$ ($F_{46}$), and $X_{487}$ ($F_{488}$)). So, 282 input variables are required.

To access all the best-obtained SEs of this research, read the Appendix A.2 subsection.

### 3.2. Classification Performance of Obtained Symbolic Expressions on the Original Dataset

In this investigation, five different dataset variations were used to train the GPSC algorithm after previous oversampling techniques were applied to the original dataset. Each dataset was used in GPSC with a 5FCV training process and for each dataset split one SE was obtained. So in this investigation, a total of 25 best SEs were obtained. These SEs were applied to the original dataset to see if these SEs could achieve the same classification accuracy as on the balanced dataset variation on which these SEs were obtained. The classification performance of all 25 SEs on the original dataset is shown in Table 6, and the confusion matrix in Figure 6.

**Table 6.** The combined classification performance of all 25 SEs applied on the original dataset.

| Evaluation Metric Name | Value |
|:---:|:---:|
| $\overline{ACC}$ | 0.9962 |
| $\sigma(ACC)$ | 0.0048 |
| $\overline{AUC}$ | 0.9949 |
| $\sigma(AUC)$ | 0.0055 |
| $\overline{Precision}$ | 0.9984 |
| $\sigma(Precision)$ | 0.0023 |
| $\overline{Recall}$ | 0.9965 |
| $\sigma(Recall)$ | 0.0056 |
| $\overline{F1-Score}$ | 0.9974 |
| $\sigma(F1-Score)$ | 0.0030 |

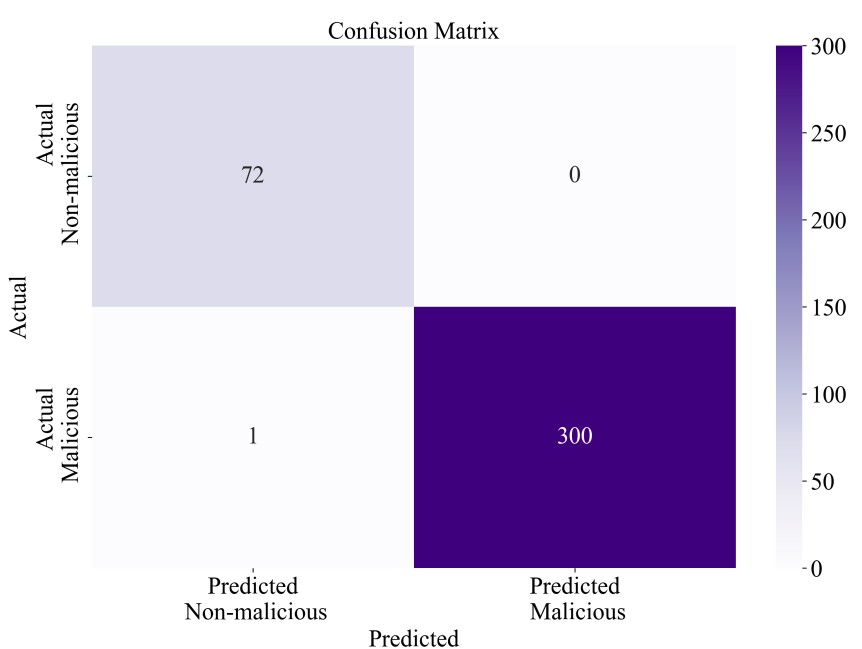

**Figure 6.** The confusion matrix for all 25 SEs applied on the original dataset.

When the classification performance of each set of SEs obtained on each dataset variation shown in Figure 5 is compared to the classification performance of all 25 SEs applied on the original imbalanced dataset in Table 6, it can be noticed that the classification

performances are similar. These results prove that the classification performance of obtained SEs on balanced datasets could be maintained when applied to the original imbalanced dataset. The confusion matrix shown in Figure 6 showed that all non_malicious exe files were classified correctly, while there was one miss-classification of the malicious class sample. So, 1/373 samples were incorrectly classified by the set of 25 SEs.

## 4. Discussion

The publicly available dataset used in this research had a great imbalance between two classes: malicious (301 samples) and non_malicious (72 samples). In this case, the class of interest had a larger number of samples than the non_malicious class. This could present a problem during the training with any machine learning algorithm, since the trained algorithm could easily detect malicious software, however it could easily misclassify the non-malicious software as malicious software. The idea was to implement various oversampling techniques, since it is generally faster to produce new synthetic data points than remove samples from the dataset as reported in [22]. So ADASYN, BorderlineSMOTE, KMeansSMOTE, SMOTE, and SVMSMOTE were chosen to synthetically achieve the balance between class samples. With the application of those oversampling techniques, five different variations of the original dataset were created. On these datasets, the GPSC was applied.

To randomly search for optimal GPSC hyperparameter values, the RHVS method was developed. The problem with this method is that it requires the initial configuration of hyperparameter boundary values and to perform the initial testing of GPSC to see GPSC performance. At this stage, the classification performance is not as important as the potential occurrence of the bloat phenomenon. So, the most challenging part in this case was to set up initial boundaries of the parsimony coefficient value, since the extremely small values can cause the occurrence of bloat phenomenon while large values (e.g., 1, 10, 100) can prevent the stable growth of population members during GPSC execution. When these initial boundaries are defined, the implementation is very simple and, before each GPSC execution, these hyperparameter values are randomly selected from its initial boundaries.

The GPSC was trained using the 5FCV process to obtain five SEs (one SE per split). With such a training procedure more robust solution is obtained. As described in the training/testing procedure, the problem is multi-objective since all mean values of the evaluation metric (ACC, AUC, Precision, Recall, and F1-Score) have to be larger than 0.99 after training. If all values are greater than 0.99, the process continues to test all five SEs on the test part of the dataset, and if the values again are greater than 0.99, the process is terminated.

Regarding the optimal combination of hyperparameters used on each dataset variation shown in Table 4, the parsimony coefficient value is small in every case (around $10^{-6}$). However, when the length of obtained SEs is analyzed (Table 5) it can be noticed that the lowest influence of parsimony coefficient was in the case of SEs obtained on a dataset balanced with the ADASYN technique, since these SEs have highest length values. The SEs obtained on the dataset balanced with the ADASYN technique achieved the highest classification performance. This performance was achieved with a lower population size when compared with BorderlineSMOTE, KMeansSMOTE, and SMOTE. The initial depth size of population members was very low when compared to other cases. As in the other four cases, the subtree mutation was the most dominating genetic operator with a 0.96 probability value. Similar values were used in other cases. The stopping criteria in all cases (for all dataset types) was low (approx. $10^{-6}$); however, it was never reached by any population member, so all GPSC executions were terminated after the maximum number of generations was reached. The range of constant values was the largest in the case of the ADASYN dataset, while in other cases the constant range was smaller.

The results obtained with the application of GPSC + RHVS and trained with 5FCV generated a set of SEs with high classification performance of each dataset balanced variation. As seen from Figure 5, the highest classification performance was achieved in the

case of SEs obtained on the dataset balanced with the ADASYN technique. Slightly lower classification performance was achieved with SEs obtained on a dataset balanced with SVMSMOTE, SMOTE, and KMeansSMOTE techniques. The SEs with the lowest classification performance were obtained on the dataset balanced with the BorderlineSMOTE technique. However, it should be noted that all evaluation metric values are in the pretty high 0.990–1.0 range.

Regarding the length and depth of obtained SEs as seen from Table 5 the lowest length and depth (62.8 and 11) were achieved with SEs obtained on the dataset balanced with the SMOTE technique. The largest SEs in terms of mean length (99.8) value are those SEs obtained on a dataset balanced with the ADASYN technique and using these SEs the highest classification performance could be achieved. The largest mean depth value was achieved in the case of SEs obtained on a dataset balanced with the SVMSMOTE technique. However the classification performance is lower than in the case of SEs obtained on the dataset balanced with ADASYN however, the length of SVMSMOTE SEs is lower.

When all 25 SEs (5 SEs per dataset variation) are applied on the original imbalanced dataset the classification performance is slightly lower than the classification performance of SEs obtained on the dataset balanced with the ADASYN technique. From Figure 6, the highest classification performance of all 25 obtained SEs can be confirmed. The SEs successfully classified all 72 non_malicious samples from the original imbalanced dataset. However, from 301 malicious samples, only 1 was misclassified. So in the entire original dataset that contains 373 samples, only 1 sample was miss-classified with the proposed method.

The final comparison of the obtained results in this research is shown in Table 7, where results from this research are compared to results from other research papers.

**Table 7.** The results obtained in this paper compared to the results from other research papers.

| Reference | ML Algorithms | Accuracy (%) |
|---|---|---|
| [11] | RFC, DNN-2,4,7 | 99.78 |
| [12] | SVM, RFC, MLP, LR, BN, AB, DT, KNN, DNN, SOM, FF, FC, DB, Y-J48, Y-SMO, Y-MLP, BTE, and MVE | 98.80 |
| [13] | LR, NB, KNN, DT, AB, RFC, SVM, CNN, MLP + SVM, and CNN + LSTM | 98.80 |
| [14] | LR, SVM, RFC | 100.00 |
| **This research** | **Dataset balancing techniques + GPSC + RHVS + 5FCV** | **99.62** |

From Table 7, it can be noticed that the research methodology in this paper produced SEs with high classification accuracy, although there is still room for improvement. The main benefit of using this approach is that after the training of GP, the SEs were obtained that can be easily used and understood unlike complex ML algorithms used in other research papers from Table 7.

## 5. Conclusions

In this paper, the GPSC with RHVS method and trained using 5FCV was applied to a publicly available dataset to develop SEs for the detection of malware software with high classification performance. The initial problem with the dataset is a large imbalance between class samples so the application of oversampling methods was mandatory to achieve a balanced dataset, which is a good starting point for the application of any AI algorithm. The RHVS method was successfully developed and applied in GPSC, and for each dataset variation, the optimal GPSC hyperparameter values were found, using which the high classification performance of obtained SEs was achieved on each dataset variation.

After SEs were obtained, they were tested on the original imbalanced dataset and the results are almost equal to those obtained on each dataset variation.

Based on the conducted investigation in this paper, the following conclusions are:

- The oversampling methods were successfully implemented and created balanced variations of the original dataset. However, in the case of ADASYN and KMeansSMOTE, the obtained versions of the dataset had an extremely small imbalance. This imbalance did not affect the classification performance of obtained SEs from these datasets.
- The application of GPSC was successful in obtaining the SEs with high classification performance in the detection of malicious software.
- Using the RHVS method, the optimal combination of GPSC hyperparameters were found for every application of GPSC on balanced dataset variation.
- A robust set of SEs with high classification performance was obtained for each dataset variation.
- Almost similar classification performance was achieved when all SEs were applied to the original imbalanced dataset, which proves that SEs were obtained with high classification accuracy. The further test revealed that only 1 out of 373 samples was misclassified.

The pros of the proposed investigation methodology are:

- With the application of oversampling methods, the balanced variations of the original dataset were created, which is a good starting point for training any AI algorithm.
- With the application of GPSC, SEs are obtained that can be easily analyzed and integrated into other software.
- The application of the RHVS method proved to be useful in finding GPSC optimal hyperparameters, using which GPSC produced SEs with high classification performance.
- The application of 5FCV produced a set of SEs that are robust

The cons of the proposed investigation are:

- It takes a long time to train the GPSC algorithm and obtain SEs using the 5FCV process, i.e., the GPSC process is repeated for each split (it is repeated 5 times).
- The initial definition and testing of GPSC boundaries in the RHVS method is a challenging process that requires multiple GPSC executions.
- The search for optimal GPSC hyperparameter values using the RHVS method can take some time

**Author Contributions:** Conceptualization, N.A. and S.B.Š.; methodology, N.A.; software, N.A. and S.B.Š.; validation, N.A. and S.B.Š.; formal analysis, N.A. and S.B.Š.; investigation, N.A. and S.B.Š.; resources, N.A.; data curation, N.A.; writing—original draft preparation, N.A.; writing—review and editing, N.A. and S.B.Š.; visualization, N.A. and S.B.Š.; supervision, N.A. and Z.C.; project administration, Z.C.; funding acquisition, Z.C. All authors have read and agreed to the published version of the manuscript.

**Funding:** This research was (partly) supported by the CEEPUS network CIII-HR-0108, the European Regional Development Fund under Grant KK.01.1.1.01.0009 (DATACROSS), the Erasmus+ project WICT under Grant 2021-1-HR01-KA220-HED-000031177, and the University of Rijeka Scientific Grants uniri-mladi-technic-22-61 and uniri-tehnic-18-275-1447.

**Data Availability Statement:** Publicly available dataset (Malware executable detection) available at: https://www.kaggle.com/datasets/piyushrumao/malware-executable-detection (accessed on 30 September 2023).

**Conflicts of Interest:** The authors declare no conflict of interest.

## Appendix A

*Appendix A.1. Modification of Mathematical Functions*

In the Materials and Methods section where GPSC was described, it was mentioned that some mathematical functions had to be modified to avoid the occurrence of Not a

number or infinite values. These functions are division, square root, natural logarithm, and logarithm with bases 2 and 10. The division function is defined as:

$$y(x_1, x_2) = \begin{cases} \frac{x_1}{x_2} & |x_2| > 0.001 \\ 1 & |x_2| < 0.001 \end{cases} \tag{A1}$$

The square root is defined as:

$$y(x) = \sqrt{|x|} \tag{A2}$$

The natural logarithm and logarithm with bases 2 and 10 are defined as:

$$y_{\log}(x) = \begin{cases} \log(|x|) & |x| > 0.001 \\ 0 & |x| < 0.001 \end{cases} \tag{A3}$$

*Appendix A.2. How to Access and Use Obtained SEs*

All the best SEs obtained on the balanced dataset variations can be downloaded from the GitHub repository: https://github.com/nandelic2022/MalwareEXEDetection.git After downloading the file simply download the dataset. (https://www.kaggle.com/datasets/piyushrumao/malware-executable-detection) and use the following procedure to calculate the output or evaluation metric values:

1. Calculate the output by providing input dataset values,
2. uses the output of each SE as the input in the Sigmoid function.
3. After the values for the samples and all SEs are obtained calculate the accuracy score, area under receiver operating characteristics curve, precision score, recall score, and f1-score using functions defined in the python scikit-learn library.

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
