# Peer review of "Improvement of Malicious Software Detection Accuracy through Genetic Programming Symbolic Classifier with Application of Dataset Oversampling Techniques"

_computers, doi:10.3390/computers12120242_

Round 1

Reviewer 1 Report

Comments and Suggestions for Authors

The references in the introduction section could be enhanced. For example, the paper mentioned that "Effective detection upholds trust, reliability, and the security of interconnected systems, preventing cyberattacks and data breaches from spreading across the digital ecosystem". This statement is true, but it would be better to cite related materials to support your statement, such as the existing work (DroidPerf: Profiling Memory Objects on Android Devices [MobiCom'23]), which detects malicious memory operations. You may also cite more existing works to show how the timely and accurate detection works in real-world cases.

In section 2.2, what's the standard for picking Kaggle as the dataset?

In section 2.3, did you cherry-pick these five oversampling techniques (ADASYN, 206 BorderlineSMOTE, KMeanSMOTE, SMOTE, and SVMSMOTE)?

To better let the readers understand the proposed five oversampling techniques, I suggest adding a table comparing them after section 2.3.5.

Please Elaborate more on each step of Figure 10, for example, the detailed workflow.

Comments on the Quality of English Language

In every bulleted list throughout the article, please check and consistently use lower or upper case for the first word (for example, between lines 75 and 83, please use "is" or "Is" consistently).

Please fix some Figure numbers, such as lines 580, 167, etc.

Author Response

The authors of this manuscript want to thank the reviewer for his comments and suggestions that have greatly improved the manuscript's quality. We sincerely hope that the revised version of the manuscript in this form can be accepted for publication. 

The references in the introduction section could be enhanced. For example, the paper mentioned that "Effective detection upholds trust, reliability, and the security of interconnected systems, preventing cyberattacks and data breaches from spreading across the digital ecosystem". This statement is true, but it would be better to cite related materials to support your statement, such as the existing work (DroidPerf: Profiling Memory Objects on Android Devices [MobiCom'23]), which detects malicious memory operations. You may also cite more existing works to show how timely and accurate detection works in real-world cases.

The authors want to thank the reviewer for his comment regarding the introduction. In the revised version of the manuscript, we have added some references. However, the sentence that the reviewer cited (if it was cited) does not exist in the manuscript. Our sentence is “Effective detection not only shields individual users and organizations from the dire consequences of these threats, such as data breaches and financial losses but also plays a pivotal role in safeguarding the broader digital landscape.”

The reference that the reviewer suggested is added to the manuscript.

The first paragraph of the introduction section did not have any references so we have included a couple of them. We have added a short description of malware detection using binary hexadecimal inspection and DLL cally analysis in the introduction section since the used dataset in this research consists of data obtained from these analyses. 

In section 2.2, what's the standard for picking Kaggle as the dataset?

The choice of Kaggle dataset was because we were searching for dataset regarding malware of malicious software. We have also searched for types of datasets that were unbalanced so thaat we could test the balancing methods and achieve high estimation accuracy on the original imblanced dataset.

In section 2.3, did you cherry-pick these five oversampling techniques (ADASYN, 206 BorderlineSMOTE, KMeanSMOTE, SMOTE, and SVMSMOTE)?

As explained in subsection 2.3 entitled “Oversampling techniques” the idea of choosing oversampling methods was twofold: 

  1. The entire dataset has only 373 samples which is a very small number of dataset samples. There are 72 samples of non-malicious and 301 malicious samples. If we had applied under-sampling techniques these techniques would have reduced the number of malicious samples and the entire dataset would have been reduced on 144 samples in total. So, the application of oversampling techniques were mandatory in order to ensure the largest possible number of samples for training of GPSC algorithm 
  2. The application of chosen oversampling techniques does not require additional fine parameter tuning. Basically with default parameters which were also used in this research the balance datasets were achieved instantly. However, if we have used the undersampling techniques these techniques would require additional fine parameter tuning. The other problem with these techniques is that even with the application of fine-parameter tuning the achieving the balanced dataset is questionable since the balanced dataset using these techniques sometimes cannot be achieved. 

The basic form of this explanation is given in the subsection 2.3. Oversampling techniques. Citing from the revised version of the manuscript: “These techniques were chosen since the original dataset has a small number of samples (373), they are faster to execute than the majority of the undersampling techniques, and there is no need to adjust its parameters as in the case of undersampling techniques.”

However, in the revised version of the manuscript, the previous explanation was expanded for a better understanding the application of oversampling techniques. Citing from the revised manuscript version in subsection 2.3. Oversampling techniques: “...These techniques were selected for three main reasons: dealing with a small dataset, ensuring faster execution, and eliminating the need for parameter tuning. The dataset used in this study is limited, with 373 samples in total, 72 in the minority (non-malicious) class, and 301 in the majority (malicious) class. Oversampling methods are preferred for quickly balancing datasets compared to under-sampling techniques like AllKNN and Edited Nearest Neighbors. For instance, AllKNN, an undersampling method, identifies and removes instances close to their k nearest neighbors in the majority class, which can be time-consuming regardless of the sample size. Another advantage of oversampling is the absence of parameter tuning, as these techniques often achieve balanced datasets without any tuning. In contrast, undersampling methods may require multiple executions with different parameters to achieve class balance. …”

To better let the readers understand the proposed five oversampling techniques, I suggest adding a table comparing them after section 2.3.5.

Another reviewer commented that this part of the descriptions of the oversampling techniques is too large and has to be shortened as much as possible. All the Figures showing PCA plots as well as unnecessary descriptions of oversampling techniques were removed. In the subsection Oversampling Techniques a short description of each oversampling technique is given as well as their pros and cons with a table just below that shows the numbers of each class after oversampling and comparing them to the original dataset samples. 

Please Elaborate more on each step of Figure 10, for example, the detailed workflow.

In Figure 10 the procedure of obtaining the best SEs with highest classification accuracy is shown with the process of random hyperparameter value search method and GPSC trained with 5-fold cross-validation. In the revised manuscript version the steps (1-5) below Figure 10 are expanded as much as possible. 

  1. The dataset (balanced dataset variation) was initially divided into training and testing datasets in a 70:30 ratio where 70% was used for training using a 5-fold cross-validation process and the remaining 30% for testing. At this stage, the GPSC hyperparameters are randomly chosen from the predefined ranges.
  2. When the dataset is split into train and test datasets and the random hyperparameter values are chosen using the RHVS method the training of the GPSC algorithm in the 5-FCV process can begin. In 5-FCV the GPSC is trained 5 times i.e. a total of 5 SEs were obtained after the 5-FCV process is completed. After the 5-FCV process is completed the accuracy, area under the receiver operating characteristics curve, precision, recall, and f1-score values are obtained for training and validation folds. Then the mean and standard deviation of the previously mentioned evaluation metrics were calculated. 
  3. All mentioned mean evaluation metric values have to be greater than 0.99 to be considered as the set of SEs with acceptable classification performance. If one of the evaluation metrics is lower than 0.99 value all the SEs are neglected and the process starts from the beginning where random hyperparameters are randomly selected and GPSC training using the 5-FCV process is repeated. However, if all evaluation metric values are higher than 0.99 the process continues to the testing phase. 
  4. In the testing phase, the remaining 30\% of the dataset is used on all 5 SEs obtained from the previous step. Using the testing set in these 5SEs the output is generated and compared to the real output to calculate the evaluation metric values. If all evaluation metric values are higher than 0.99 then the process is completed. However, if one evaluation metric value is lower than 0.99 the process starts from begging by randomly selecting the hyperparameter values. 

Comments on the Quality of English Language

In every bulleted list throughout the article, please check and consistently use lower or upper case for the first word (for example, between lines 75 and 83, please use "is" or "Is" consistently).

The inconsistencies in the bulleted lists are corrected throughout the manuscript. Thank you for pointing that out. 

Please fix some Figure numbers, such as lines 580, 167, etc.

The  Figure numbers are fixed throughout the manuscript. Thank you for pointing that out. 

Reviewer 2 Report

Comments and Suggestions for Authors

The paper focuses on addressing high imbalance datasets using oversampling techniques. The authors have developed a random hyperparameter value search method to find the optimal hyperparameter values for a GPSC classifier. However, there are several points that require clarification and improvement in the paper:

  1. The paper does not provide clear information about the DDL in the abstract.
  2. The authors should ensure that the figure number format is consistent throughout the paper. Specifically line 167, 580
  3. It is not advisable to use the test set results to adapt the random hyperparameter values search method. This may lead to overfitting and undermine the validity of the results. Improved experimental design is needed.
  4. The authors should add the oversampling step to the training/testing procedure figure (Figure 10). And this step should ideally be performed on the training set separately after splitting the test/training set. This adjustment would more accurately reflect the workflow.
  5. The content between lines 479-515 should be simplified. The material seems to contain unnecessary and meaningless variables. Please ensure that the information presented is relevant and contributes to the paper's clarity.
  6. The paper should explicitly state its novel contributions. It is important to clearly convey what sets this research apart from existing work.

Overall, the paper addresses an important issue related to high imbalance datasets and the optimization of hyperparameter values. However, there are some concerns related to clarity, figure formatting, method design and the presentation of results. Additionally, it is crucial to emphasize the novelty of the research.

Comments on the Quality of English Language

Please also check minor grammar issues for enhanced readability

Author Response

The authors of this manuscript want to thank the reviewer for his comments and suggestions that have greatly improved the manuscript's quality. We sincerely hope that the revised version of the manuscript in this form can be accepted for publication. 

The paper focuses on addressing high imbalance datasets using oversampling techniques. The authors have developed a random hyperparameter value search method to find the optimal hyperparameter values for a GPSC classifier. However, there are several points that require clarification and improvement in the paper:

  • The paper does not provide clear information about the DDL in the abstract.

The authors of this paper want to know what is DDL. We have not found the DDL in our paper. Did you mean DLL? If so, the authors think that readers who would eventually read the manuscript are familiar with the DLL. The information on DLL is given in the introduction section and again before the dataset description. Generally, the available information online of the used dataset in this paper is very short so we provided information about DLL as best as we could. 

  • The authors should ensure that the figure number format is consistent throughout the paper. Specifically line 167, 580

In the revised version of the manuscript, this was corrected. Thank you for noticing these mistakes.

  • It is not advisable to use the test set results to adapt the random hyperparameter values search method. This may lead to overfitting and undermine the validity of the results. Improved experimental design is needed.

The test dataset does not have any connection with the random hyperparameter search method. First of all the random hyperparameters are selected before training using 5FCV. After 5FCV was completed 5 SEs were obtained and evaluation metric values (for all 5 Ses including mean and standard deviation values) were obtained. If all evaluation metric values are greater than 0.99 then these 5-SEs are tested on the testing dataset (remaining 30\%). If evaluation metric values on the testing dataset for all 5 Ses are greater than 0.99 then the process is completed. Please check the results (Figure 11) the overfitting did not occur.

  • The authors should add the oversampling step to the training/testing procedure figure (Figure 10). And this step should ideally be performed on the training set separately after splitting the test/training set. This adjustment would more accurately reflect the workflow.

We do not need to add the oversampling step since this was done before. Figure 10 shows the procedure of how the balanced dataset variation is used in GPSC to obtain symbolic expressions in a 5-fold CV process with the GPSC random hyperparameter value search method.  To clarify this misleading Figure in the revised version of the manuscript the term 

  • The content between lines 479-515 should be simplified. The material seems to contain unnecessary and meaningless variables. Please ensure that the information presented is relevant and contributes to the paper's clarity.

The authors agree with the reviewer’s comment. The idea was to show all necessary input variables needed to compute the output. Investigation showed that the majority of variables are necessary to compute the output. In the revised version of the manuscript, we have eliminated some of the required variables from the description. For example, instead of showing X_59, X_60, X_61, X_62, and X_63, we have shown these variables in the following form X_59 - X_63. By doing so we have reduced this description from the initial line range 479 - 515 to 456-483. The lower line boundary changed due to other modifications made to our manuscript.

  • The paper should explicitly state its novel contributions. It is important to clearly convey what sets this research apart from existing work.

In this paper, the novelties are:

  1. Procedure to obtain a highly accurate and robust set of SE ensemble using genetic programming symbolic classifier for the detection of malicious software.   
  2. Procedure how to easily achieve a balanced dataset without any parameter tuning (oversampling techniques) and use this balanced dataset to obtain highly accurate symbolic expressions (SEs).

The validity of our approach is shown at the end of the manuscript (confusion matrix) where we have achieved high classification accuracy using an obtained set of SEs on the original imbalance dataset.

Another benefit of utilizing this approach is the ability to obtain symbolic expressions which can in future use be easily integrated for the detection of malicious software. 

The idea and novelty where emphasized before defined hypotheses in the introduction section and again in the conclusions section. 

Overall, the paper addresses an important issue related to high-imbalance datasets and the optimization of hyperparameter values. However, there are some concerns related to clarity, figure formatting, method design and the presentation of results. Additionally, it is crucial to emphasize the novelty of the research.

We have emphasized the novelty of this paper throughout the entire paper. To the best of our knowledge, this is the in which symbolic expressions were obtained for malicious software detection. The ability to obtain the mathematical expression in which malware software detection can be done with high accuracy is a novelty. When compared to other machine learning and deep learning methods the classification accuracy is almost 100%. As stated in the introduction section the benefit of using obtained mathematical equations is that they require low computation resources to execute when compared to complex machine or deep learning models. The other novelty is the creation of a symbols expression ensemble consisting of the large number of symbolic expressions that were obtained on balanced dataset variations. This system is highly accurate and robust since the final evaluation of the imbalance dataset showed i.e. that high accuracy was achieved.

Reviewer 3 Report

Comments and Suggestions for Authors

- When indicating accuracy in Table 1, the number of decimal places must be unified.

- In line 167, Figure ?? must be numbered.

- It would be a good idea to indent line 179.

- In Table 6, the value must be written with the correct number of significant figures.

- It would be a good idea to indent lines 547 and 587.

- In Table 7, accuracy must be indicated with significant figures.

Overall, I think it is a well-written paper. However, it would be good to organize the content by eliminating some of the unnecessary parts and emphasizing the parts that are absolutely necessary for research. For example, section 2 is too long, but it would be better to briefly compare each oversampling technique.

Comments on the Quality of English Language

n/a

Author Response

The authors of this manuscript want to thank the reviewer for his comments and suggestions that have greatly improved the manuscript's quality. We sincerely hope that the revised version of the manuscript in this form can be accepted for publication. 

- When indicating accuracy in Table 1, the number of decimal places must be unified.

In the revised manuscript version the number of decimal places is unified to two decimal places. 

- In line 167, Figure ?? must be numbered.

The authors want to thank the reviewer for noticing the mistake. The sentence in which these two ?? signs appeared was also unfinished. So we have completed the sentence and put the number after the Figure. 

- It would be a good idea to indent line 179.

No problem. In the revised version of the manuscript, line 179 is indented by 0.25 cm. 

- In Table 6, the value must be written with the correct number of significant figures.

In the revised manuscript version all values were written up to 4 decimal places. These 4 decimal places are enough to show significant figures.

- It would be a good idea to indent lines 547 and 587.

Lines 547 and 587 are indented in the revised manuscript version. 

- In Table 7, accuracy must be indicated with significant figures.

In Table 7 the number of decimal places is unified and the results of our research are bolded. 

Overall, I think it is a well-written paper. However, it would be good to organize the content by eliminating some of the unnecessary parts and emphasizing the parts that are absolutely necessary for research. For example, section 2 is too long, but it would be better to briefly compare each oversampling technique.

Section 2 is reduced by eliminating PCA plots and a description of each oversampling technique.  The PCA is unnecessary since the minority class that is oversampled is concentrated on an extremely small region in these plots so it is hard to notice the synthetically generated samples. This subsection (oversampling techniques) was reduced to one page where a short description of each oversampling technique was given as well as pros and cons and finally, the table in which the results of the application of oversampling techniques on the original dataset are listed.

Round 2

Reviewer 2 Report

Comments and Suggestions for Authors

Accept in present form